# Nanoparticle-Based Bivalent Swine Influenza Virus Vaccine Induces Enhanced Immunity and Effective Protection against Drifted H1N1 and H3N2 Viruses in Mice

**DOI:** 10.3390/v14112443

**Published:** 2022-11-03

**Authors:** Pan Tang, En-hui Cui, Wen-chi Chang, Chen Yu, Hao Wang, En-qi Du, Jing-yu Wang

**Affiliations:** 1College of Veterinary Medicine, Northwest A&F University, Yangling, Xianyang 712100, China; 2Yangling Carey Biotechnology Co., Ltd., Yangling, Xianyang 712100, China

**Keywords:** nanoparticle, vaccine, protective immunity, adjuvant, swine influenza virus

## Abstract

Swine influenza virus (SIV) circulates worldwide, posing substantial economic loss and disease burden to humans and animals. Vaccination remains the most effective way to prevent SIV infection and transmission. In this study, we evaluated the protective efficacy of a recombinant, baculovirus-insect cell system-expressed bivalent nanoparticle SIV vaccine in mice challenged with drifted swine influenza H1N1 and H3N2 viruses. After a prime-boost immunization, the bivalent nanoparticle vaccine (BNV) induced high levels of hemagglutination inhibition (HAI) antibodies, virus-neutralization (VN) antibodies, and antigen-specific IgG antibodies in mice, as well as more efficient cytokine levels. The MF59 and CPG1 adjuvant could significantly promote both humoral and cellular immunity of BNV. The MF59 adjuvant showed a balanced Th1/Th2 immune response, and the CPG1 adjuvant tended to show a Th1-favored response. The BALB/c challenge test showed that BNV could significantly reduce lung viral loads and feces viral shedding, and showed fewer lung pathological lesions than those in PBS and inactivated vaccine groups. These results suggest that this novel bivalent nanoparticle swine influenza vaccine can be used as an efficacious vaccine candidate to induce robust immunity and provide broad protection against drifted subtypes in mice. Immune efficacy in pigs needs to be further evaluated.

## 1. Introduction

Influenza viruses are prevalent worldwide and can infect humans and various animals [1,2]. Pigs play a complex and vital role in the interspecies transmission of influenza viruses [3,4]. The well-known pandemic H1N1 influenza virus (pdm/09), also known as the ‘swine flu,’ was first identified in Mexico in April 2009 [5]. Swine influenza virus (SIV) poses a potential economic burden to the pig industry and causes serious harm to public health [6]. Clinically, SIV infection tends to have low mortality, but it often causes progressive wasting, poor growth performance, and may cause severe pneumonia or even death due to co-infection with other porcine respiratory pathogens or bacterial diseases [7,8]. Vaccination of pigs remains the most effective and economical strategy for preventing SIV infections and reducing the severity of the infection and its severe complications.

SIV evolved in two major mechanisms: antigenic drift and antigenic shift [9]. Antigenic drift frequently occurs due to the accumulation of point mutations of HA and NA during virus evolution [10,11]. Antigenic shift occurs rarely, usually accompanied by the generation of a completely new virus by the genetic reassortment of the multiple gene segments [12]. Eventually, these changes are introduced into the circulating strains and can lead to epidemics (antigenic drift) and pandemics (antigenic shift) of influenza viruses, which also cause decreased vaccination effectiveness and periodic antigenic revision for the SIV vaccine [13].

Classical swine H1N1, avian-like H1N1, and humanoid H3N2 strains are widely prevalent in pigs [14]. Serological epidemiological investigation of swine influenza (SI) shows that the distribution and evolution of SI in pig herds in China are becoming increasingly complex [15,16,17,18,19,20,21].

Hemagglutinin (HA) is one of the main surface glycoproteins of influenza virus and plays a critical role in the early stages of virus infection. The transmembrane (TM) region of the HA protein of the H3 subtype influenza virus can form intermolecular disulfide bonds, which makes HA protein have high stability and cross-immunity between subtypes [22,23,24,25]. 

Currently, most commercial influenza vaccines are manufactured using embryonated chicken eggs. However, the adaptive mutations of the influenza virus during serial passages often weaken vaccine effectiveness [26]. Nowadays, MDCK and VERO cell-based influenza vaccine production have become an important alternative to conventional egg-based production systems and have received regulatory approval in some countries [27,28,29]. Baculovirus-based seasonal influenza vaccines have also completed phase III clinical trials [30]. The conventional influenza vaccine cannot protect against antigenic-drifted or antigenic-shifted strains of SIV, and novel vaccines that induce broadly immune responses are needed.

Nanoparticle vaccines can exhibit multiple viral antigen components on specific nanoskeleton surfaces and stimulate strong humoral and cellular immune responses. It has shown broad application prospects in preventing and treating infectious or serious diseases such as COVID-19, hepatitis B, influenza, AIDS, and tumors [31,32,33,34,35,36].

In this report, we developed a recombinant, baculovirus-insect cell system-expressed, HA TM region-replaced nanoparticle with a skeleton core of PS80 micelle for bivalent swine influenza vaccine. After a prime-boost immunization, the immunogenicity and protective efficacies of the bivalent nanoparticle vaccine (BNV) against drifted SIVs in mice were evaluated.

## 2. Materials and Methods

### 2.1. Cloning and Expression of HA Nanoparticles (HANP)

SIV A/swine/Qingdao/2018 (H1N1) and A/swine/Jiangsu/P3589/2016 (H3N2) HA protein sequences were downloaded from the GISAID Epiflu database with accession numbers EPI_ISL_370660 and EPI_ISL_256394. 

For the recombinant H1N1 strain, the wild-type (WT) HA TM domain was replaced with the H3-TM domain (Figure 1a). All HA genes were codon-optimized for high-level expression in Spodoptera frugiperda (Sf9) insect cells (Sf-900™, Cat. No. 12659017, Gibco, Carlsbad, USA) and synthesized biochemically by Genscript (Nanjing, China). HA genes were cloned into pBacPAK9 baculovirus transfer vectors between BamHI-KpnI sites downstream from a polyhedrin promoter. Furthermore, pBacPAK9 plasmids containing each HA gene were co-transfected into Sf9 cells with qBac Bacmid containing Autographa Californica multinuclear polyhedrosis virus genome (Shaanxi Bacmid Biotechnology Co., Ltd., Yangling, China) and SofastTM Transfection reagent (Sunma, Xiamen, China).

### 2.2. Cell Lines

Sf9 cells (Gibco™ Sf-900™ III SFM, Gibco™ 12659017, Cat. No. 15236025, Carlsbad, CA, USA) and high five cells (BTI-TN-5B1-4, Invitrogen™ B85502, Cat. No. 10747474, Carlsbad, CA, USA) were purchased by Xi’an Lihe Biotechnology Co., LTD (Xi’an, China) from Thermo Fisher Scientific (Waltham, MA, USA). Sf9 cells were used to isolate and propagate recombinant baculoviral stocks, and high five cells were used to express the recombinant protein. All cells were cultured in IB905 insect serum-free medium (world-medium, Suzhou, China).

### 2.3. Virus Titration

Virus titers were determined using a BacPAK™ Baculovirus Rapid Titer Kit (Clontech) according to the manual (https://www.takarabio.com/documents/User%20Manual/PT3153/PT3153-1_072213.pdf, accessed on 16 April 2022). The foci of infection (clusters of infected cells) were counted in duplicate wells using an inversion microscope (CKX31SF, Olympus, Tokyo, Japan). Virus titers (IFU/mL) were calculated by multiplying the average number of foci per well by the corresponding dilution factors.

### 2.4. Hemagglutination Assay

Hemagglutination assays were determined according to the manual for GB/T-27536-2011 of the state standard of the People’s Republic of China. In brief, 50 µL of PBS was added to wells of each row of 96-well microtiter plates, 50 µL of the recombinant baculovirus suspension was then serially diluted from 1–11 columns of the plate, and 50 µL was discarded from the 11th column. The last row of the plate served as negative controls. After dilution, 50 µL of the 0.5% chicken erythrocytes suspension was added to each well, then the plate was incubated at 25 °C for 30 min. The last well with visible signs of hemagglutination was determined as the hemagglutination titer.

### 2.5. Purification of Recombinant HANP

The purification of HA proteins was performed as previously described. Briefly, Sf9 cells were maintained as suspension cultures in SF-SFM insect serum-free medium (world-medium, Suzhou, China) at 27 °C of 110 rpm. Sf9 cell cultures were infected with recombinant baculovirus (rBV) expressing HA genes, and infected cells were harvested by centrifugation at 10,000 g for 30 min. HA was extracted from cell membranes with non-ionic detergent Sango^®^ Triton X-100 and purified with ion-exchange chromatography, affinity chromatography, tangential flow, and aseptic filtration. Detergent exchange to polysorbate 80 (PS80) was performed during chromatography forming HANP. The nanoparticle was formed by combining the HA trimers hydrophobic transmembrane regions with the PS80 micelle. 

### 2.6. TEM and DLS

Transmission electron microscopy (TEM) images of the HANP were obtained by negative staining. The purified HANP sample was diluted to 10 µg/mL in formulation buffer without PS80. The HANP sample (3 µL) was applied to a nitrocellulose-supported 400-mesh copper grid for several seconds, washed with Milli-Q water, and negatively stained with 1% phosphotungstic acid. Upon drying, images were recorded on a Tecnai G2 microscope (FEI) at 80 kV with a CCD camera (Morada G3).

The hydrodynamic diameter (Z-average size) of influenza HANP protein was measured at 25 °C by dynamic light scattering (DLS) using a Zetasizer Nano ZS instrument (Malvern instruments limited., Worcestershire, UK). Particles were dispersed using sterile phosphate-buffered saline at a concentration of 1 mg/mL before determination of size/PDI. The Z-average is represented as diameter in nm ± width of the distribution.

### 2.7. Vaccine and Adjuvant

BNV consisted of a dose of 5 µg HA/strain of A/swine/Qingdao/2018 (H1N1) and A/swine/Jiangsu/P3589/2016 (H3N2) in 0.2 mL, respectively. BNV was co-administered with 5 µg MF59 or 5 µg CPG1 adjuvant (5′-TsCsGsCsCsCsGsTsCsGsCsCsAsGsCsGsAsGsGsCsGsTsTsT-3′, Yiweikang Bio, Wuhan, China), separately. Commercial bivalent inactivated swine influenza vaccine was purchased from Sinovet Bio (Taizhou, China), and contains strains of A/Swine/Liaoning/32/2006 (H1N1) and A/Swine/Heilongjiang/10/2007 (H3N2).

### 2.8. Viruses

Swine virus strains H1N1 (CVCC AV1523) and H3N2 (CVCC AV1520) were obtained from the Centers for Veterinary Culture Collection of China (China Institute of Veterinary Drug Control, Beijing, China). Virus strains used for the challenge assay were grown in pathogen-free, 10-day-old embryonated chicken eggs (Beijing Boehringer Ingelheim Vital Biotechnology Co., Ltd., Beijing, China); Viruses used for the HAI assay were grown and passaged in the high five cell.

### 2.9. Animals, Immunization, and Viral Challenge

Six-week-old female specific pathogen-free BALB/c mice were purchased from Beijing Vital River Laboratory Animal Technology Co., Ltd (Beijing, China). Mice were maintained in individually ventilated cage (IVC) systems (Fengshi Group, Suzhou, China) and supplied with ad libitum feed and water. Each group consisted of 8 mice. The room temperature was maintained at 22 °C ± 3 °C, relative humidity 50 ± 10%, and the light cycle was a 12 h on/off photoperiod. 

Balb/c mice were immunized subcutaneously with purified bivalent swine influenza nanoparticle vaccines based on HA of 5 µg and with or without MF59 or CPG1 adjuvant at weeks 0 and 4. The commercial swine influenza inactivated vaccine group was immunized with the same hemagglutination unit of antigen in 0.2 mL at week 1 and week 4, and the PBS group was set as a negative control (NC). Sera samples were obtained from all mice before vaccination and interval weeks after the first and booster immunization. Mice were challenged intranasally (i.n.) with 10^6.5^ median embryo infectious dose (EID50) of swine influenza H1N1 and H3N2 strains 0.1 mL at week 6, respectively. Following the challenge, Balb/c mice were monitored twice daily for clinical signs and weighed daily for 14 days. Mice showing >25% of their initial body weight loss were defined as reaching the experimental end-point and were euthanized humanely. Three mice of each immunized group were euthanized after 6 days post-infection, left lungs were harvested and homogenized for RNA extraction, and the right lung was fixed in 10% buffered formalin and stained with hematoxylin and eosin for histopathological examination. Feces samples were obtained before vaccination and challenge and daily after challenge until the end of the study to determine individual influenza viral RNA shedding profiles by RT-PCR according to the China National Standards (CNS) GB/T 27521-2011 developed by the Harbin Veterinary Research Institute of the Chinese Academy of Agricultural Sciences.

### 2.10. HAI and VN Assay

HAI antibody titers were measured using 0.5% chicken erythrocytes and 4 HA units of 2 recombinant influenza viruses. Prior to HAI testing, the serum specimens were treated with receptor-destroying enzyme II (RDE II, Denka Seiken Co., Ltd., Tokyo, Japan) at 37 °C overnight followed by 56 °C for 30 min to remove nonspecific hemagglutinin inhibitors and natural serum agglutinins. The HAI assays were finally performed in 96-well microtiter plates according to the manual for GB/T-27535-2011 of the state standard of the People’s Republic of China.

Virus neutralizing activity were determined using a modified neutralization assay from a previously described procedure [37]. In brief, serum samples were heat-inactivated at 56 °C for 30 min, 2-fold serial diluted and incubated with 100 TCID50/well of swine influenza virus in 96-well microplates. The mixture was incubated at 37 °C for 1 h, the serum-swine influenza virus mixture was then transferred to 96-well microplates containing MDCK cells and incubated for 72 h. Anti-swine influenza VN antibodies were measured using a standard cytopathic (CPE) inhibition, neutralization titers are expressed as the reciprocal value of the highest dilution of the serum which inhibited the production of CPE.

### 2.11. Virus-Specific IgG ELISA and Cytokine Analysis

Antigen-specific IgG and IgG subclass (IgG1 and IgG2a) levels in serum were performed using an enzyme-linked immunosorbent assay (ELISA) described previously [37]. The measurement was performed using a microplate analyzer (Prolong, DNM-9602; Prolong New Technology Co., LTD., Beijing, China) at 450 nm. Concentrations of IFN-γ and IL-4 in mouse serum samples were analyzed two weeks after booster immunization by absorbance changes at a wavelength of 450 nm by ELISA using a mouse IFN-γ/IL-4 enzyme immunoassay kit (MlBio, Shanghai, China) according to the manufacturer’s instructions.

### 2.12. Histopathology

After 6 days of the challenge, mice were euthanized, and lung tissues were collected. Lung tissues were fixed in 10% formalin, embedded in paraffin, and used for HE staining and histopathological analysis. The tissue sections cut at 4 µm were affixed on glass slides and then viewed under the Pannoramic 250 digital slice scanner (3DHistech, Budapest, Hungary).

### 2.13. Statistical Analysis

GraphPad Prism software version 9.0 was used for all statistical analyses. Statistically significant differences between experimental groups were determined by an unpaired *t*-test. *p* values (*p*) of <0.05 were considered statistically significant, while *p* < 0.01 was considered a highly significant difference. All data were reported as the mean ± standard deviation (SD).

### 2.14. Ethics Statement

All animal experiments were performed according to the guidelines of the Northwest A&F University, using protocols approved by the institutional laboratory of Animal Care and Use Committee (No. 220412). The 3R principle was strictly observed during the experiment to ensure animal welfare.

## 3. Results

### 3.1. Generation and Characterization of an rBV Expressing the H1 and H3 HA Gene

A schematic diagram of a TM-replaced rBV strain, designated as H1-TM, is shown in Figure 1a. The presence of the HA gene was confirmed by PCR and sequence analysis. The rBV plasmid pBacPAK9-H1-HA-TM and pBacPAK9-H3-HA-WT containing the HA gene of the A/swine/Qingdao/2018 (H1N1) and A/swine/Jiangsu/P3589/2016 (H3N2) virus, and the bacmid plasmid was prepared and used to co-transfect Sf9 cells. After six days of co-transfection, Sf9 cells became tumefacient and rounded, and rBV-H1-TM and rBV-H3-WT were obtained by serial passage on Sf9 cells and then expressed on Hi5 cells.

To investigate the impact of the replacement of the TM of H1-HA with an H3-TM on recombinant protein expression and hemagglutinating activity, Western blots of purified HA protein and hemagglutination assay were analyzed. Results showed that viral titration of H1-TM in Sf9 cells was higher than that of H1-WT which was determined using a BacPAK™ Baculovirus Rapid Titer Kit (Figure 1b). Western blotting showed that the HA protein expressed by rBV-H1-TM, rBV-H1-WT, and rBV-H3-WT was specifically recognized by the anti-His tag primary antibody (66005-1-Ig, Proteintech, Wuhan, China) (Figure 1c). The hemagglutination assay showed that the hemagglutination titer of H1-TM was one titer higher than H1-WT (Figure 1d).

### 3.2. HANP Production and Characterization

This novel HANP is a mixed micelle consisting of recombinantly expressed full-length HA trimers with an intact membrane domain which stably interacts with a core of PS80 molecules. The purity of the HANP protein was determined by SDS-PAGE densitometry, and it was >90%. The particle size mostly ranged from 32.67 to 68.06 nm in diameter as determined using DLS (Figure 2a). The nanoparticle structures were characterized by electron microscope negatively stained images exhibiting variable HA trimers per nanoparticle. The result is consistent with previous reports [38] (Figure 2b).

### 3.3. Humoral and Cellular Immune Responses Elicited by BNV

The immunization and challenge of mice were carried out according to the experiment scheme (Figure 3a). To assess the immunogenicity of the BNV, mice were immunized twice with BNV with or without MF59 or CPG1 adjuvant, and serum samples were collected from all mice after the second immunizations at week six. Then serological HAI and VN antibody titers were measured and rBV supernatants expressed on Hi5 cells were used as antigens in the HAI test [39]. 

Our results showed that no HAI antibodies were detected from groups inoculated with PBS, whereas high HAI titers were detected in the BNV group (Figure 3b,c). In a comparison of BNV against inactivated influenza vaccines (IIV) in mice, HAI titers induced by BNV were four-fold higher than IIV in the absence of the adjuvant. MF59 and CPG1 adjuvants can remarkably enhance the HAI antibody titers of BNV, and the average HAI titer of the MF59 adjuvant group was two to four times that of the single nanoparticle group. There was no significant difference between the two adjuvants groups. A similar effect was observed in the VN antibody titers of H3N2 (Figure 3e), whereas the VN antibody titers between the two adjuvants groups of H1N1 showed significant difference (Figure 3d). The VN antibody titers from the group immunized with BNV against H1N1 and H3N2 arrived at the highest level of 2.7 log10 and 2.4 log10 in the 6th week, whereas the VN antibody titers of the IIV group reached only 2.1 log10 and 1.8 log10. Furthermore, the VN antibody titers against H1N1 virus of MF59 and CPG1 adjuvant groups arrived at the highest level of 3.6 log10 and 3.9 log10, respectively, and H3N2 of 3.3 log10 and 3.6 log10.

Antigen-specific IgG1 and IgG2a levels in serum were correlated with the magnitudes of Th2 and Th1 responses, respectively. Two weeks after the boost immunization, antigen-specific IgG1 and IgG2a levels in serum were measured by ELISA. MF59 and CPG1 enhanced H1N1/H3N2-HA-specific IgG and IgG subclass responses. BNV combined with either MF59 or CPG1 produced more significant IgG and IgG subclass antibodies than BNV and IIV groups (Figure 4a–d), and the level of IgG1 IIV group was higher than that of IgG2a, indicating Th2-favored responses. IgG1 and IgG2a antibodies were comparable in the BNV alone and MF59 + BNV groups and showed a balanced immune response, whereas the CPG1 + BNV group showed a Th1-favored responses. 

Cellular immunity plays an important role in preventing influenza virus infection. The production of cytokines in vaccination determines the magnitude and dimension of immune responses. Concentrations of samples were analyzed two weeks after booster immunization by absorbance changes at a wavelength of 450 nm by ELISA. IFN-γ and IL-4 levels in serum were evaluated two weeks after boosting immunization. As shown in Figure 4e,f, IFN-γ levels in BNV alone and MF59- or CPG1-adjuvanted BNV groups increases compared with the PBS or IIV groups. A similar trend was observed in the IL-4 test, with no significant difference between MF59 + BNV and CPG1 + BNV groups.

Taken together, BNV can elicit high and balanced humoral and cellular immune responses, mainly cellular immunity in the presence of the CPG1 adjuvant.

### 3.4. Protective Efficacy against Drifted H1N1 and H3N2 SIV Infection

We then determined the protection level of BNV against homologous but drifted virus attacks. Protein blast results showed that the amino acids homology of HA protein between the vaccine strains and the drift challenge strains used in this study were quite different, with A/swine/Qingdao/2018 (H1N1) and CVCC AV1523 (H1N1) at 81.8%, and A/swine/Jiangsu/P3589/2016 (H3N2) and CVCC AV1520 (H3N2) at 80.39%, and they belong to different branches.

At 14 days after the booster immunization, vaccinated and control mice were challenged with drifted SIV H1N1 (CVCC AV1523) and H3N2 (CVCC AV1520), respectively. The protective efficacy was evaluated by the mental state, weight loss, viral loads in lungs, and viral shedding in feces of the mice following the stringent challenge. A slight weight loss in mice was observed after challenge in the BNV-vaccinated and IIV-vaccinated group, while the mice in the PBS-inoculated group showed weight losses of 27.2% and 17.5%, respectively, one week after the challenge (Figure 5a,b). Although weight gain was observed on subsequent days, the body weight of the PBS group still did not recover to the pre-challenge level by day 14. The survival rates of mice were recorded over time, and survival rates were illustrated using the Kaplan–Meier method (Figure 5c,d).

At six days post-challenge (dpc), mice immunized with BNV plus MF59 and CPG1 showed no lung viral loads following the challenge with H1N1 and H3N2 viruses. Furthermore, viruses collected from mice inoculated with BNV showed a significantly lower lung viral load than those in the IIV group (*p* < 0.0001). In contrast, mice inoculated with the PBS group showed high loads of infectious virus in lung tissues, with viral loads of H1N1 of 6.4 × 10^8^ copies/mL, and H3N2 of 6.8 × 10^7.6^ copies/mL, respectively (Figure 5e,f).

Compared to control groups with BNV-vaccinated groups of viral shedding in feces by RT-PCR, BNV-associated groups of mice displayed a significant reduction in viral shedding on days four and six dpc (Figure 5g,h). The addition of MF59 and CPG1 adjuvants to the BNV vaccine provided enhanced protection against virus shedding compared with BNV alone, and the virus was undetectable in feces after 14 days of the challenge.

These results indicate that this novel swine influenza BNV could provide complete protection against drifted swine influenza H1N1 and H3N2 viruses.

### 3.5. BNV Reduces Lung Pathological Lesions upon Homologous SIV Challenge

The mice of each group were challenged with the H1N1 (CVCC AV1523) and H3N2 (CVCC AV1520) viruses, and the lung tissues of mice were collected at six dpc for pathological examination. Lungs of unimmunized mice are shown in Figure 6A and Figure 7A. The structure of lung tissues in mice vaccinated with IIV were basically normal, only slight histopathologic changes were found (Figure 6B and Figure 7B). No obvious pathological changes were observed in mice vaccinated with BNV and BNV plus MF59 or CPG1 adjuvant groups (Figure 6C–E and Figure 7C–E). In contrast, the PBS-vaccinated group showed alveolar wall congestion, disappeared alveolar cavity structure, alveolar epithelial cell necrosis, cell structure disintegration, nuclear pyknosis, and dissolution, accompanied by the slight proliferation of alveolar epithelial cells and more inflammatory cells were infiltrated. Mainly round and deeply stained lymphocytes showed typical symptoms of interstitial pneumonia after challenging with H1N1 (CVCC AV1523) and H3N2 (CVCC AV1520) viruses (Figure 6G and Figure 7G). There were no obvious symptoms of tracheal hemorrhage in immune groups, only mild tracheal wall thickening and lymphocytic infiltration were observed (Figure 6H,I and Figure 7H,I). Generally, according to the clinical symptoms and pathological changes, the pathological damage caused by H1N1 was more serious than that caused by H3N2.

Taken together, these data indicated that BNV could provide effective efficacy in reducing lung gross and microscopic lesions of H1N1 (CVCC AV1523)- and H3N2 (CVCC AV1520)-challenged mice.

## 4. Discussion

Currently, there are only very limited reports on the production of the swine influenza vaccine using an insect baculovirus expression vector system (IBEVS) [40,41]. Our study shows that the recombinant bivalent swine influenza nanoparticle vaccine expressed by IBEVS has good immunogenicity and full protection against drifted SIVs, and could be chosen as a vaccine candidate during a pandemic.

Normally, it takes an average of one embryonated egg to produce one vaccine dose. The nanoparticle vaccine in this study offers many advantages, including genetic stability, high titers, as well as ease of scale-up rapidly without concerns about the shortage of embryonated eggs. Therefore, BNV appears to be more economical and readily marketable.

Previous studies showed that substitutions of cysteines in the HA TM domain and a replacement with the H3-HA TM domain could enhance stability and hetero-protection in mice [22,23,24,25]. In this study, we generated a recombinant H1N1 WT strain (H1-WT) and a recombinant H1N1 strain with an H3-TM domain replacement (H1-TM) utilizing a baculovirus expression system. The comparison of biological characteristics between the two H1-viruses indicated that the replacement of the TM domain did not affect the production of rBV and showed higher hemagglutination activity and viral titer, which is consistent with previous reports [23]. In addition, stability experiments of purified protein stored at 4 °C for one month were verified by BCA quantitative assay. The results showed that H1-TM displayed high stability, while H1-WT showed a certain extent of degradation (data not shown). Stability experiments with longer storage periods are ongoing. Previous studies also revealed that the HA proteins with replaced H3-WT TM exhibited enhanced hetero-protection [23]. Whether the research generated by the bivalent HANP swine influenza vaccine can resist attacks of heterologous virus needs to be further proven.

Nanoparticle vaccine has shown its broad biomedical application prospects in preventing and treating major human infectious diseases, such as COVID-19, hepatitis B, influenza, AIDS, and tumors [31,33,36,37,42,43]. In this study, we report a novel nanoparticle HA/PS80, a mixed micelle consisting of recombinantly expressed full-length influenza HA trimers that stably interact with the core of PS80 molecules through the hydrophobic TM domain. In terms of morphology, the HA/PS80 nanoparticles in this study showed diversity in size and shape, which was different from the familiar spherical or cage-like nanoparticles, such as self-assembling ferritin nanoparticles, two-component I53-50A/50B nanoparticles, and liposome nanoparticles [32,44,45]. DLS showed that the particle size of HA/PS80 nanoparticles in this study mostly ranged from 28.67 to 68.06 nm in diameter, which was different from previous reports on the seasonal influenza vaccine of 28.9 to 33.7 nm [38]. The result may attribute to the varying number of HA trimers bound to the PS80 micelle. Typically, 2-7 HA trimers can be bound to a PS80 micelle. Similar protein-detergent-core nanoparticles have been applied in developing a quadrivalent seasonal influenza vaccine and an RSV vaccine previously [46]. Concerning the type of immune response and efficiency, HA/PS80 nanoparticles induced a high level of balanced humoral and cellular immunity compared with inactivated vaccines, while other nanoparticles described above elicited mainly cellular immune responses. In the practice of vaccine research, whether to pursue a high level of cellular immunity or a balanced cellular and humoral immunity is a topic worthy of continuous discussion.

In this study, we also compared the enhancement effect of MF59 and CPG1 adjuvant on nanoparticle vaccines. The results indicate that MF59 adjuvant stimulates higher and stronger humoral and cellular immunity and shows milder pulmonary pathological lesions and lower viral load and fecal shedding in challenge protection studies than inactivated and single nanoparticle vaccines. CPG1 adjuvant groups mainly induce cellular immunity, and the level of IgG2a antibody, which symbolizes cellular immunity, is more than 1.6 times that of IgG1. Furthermore, CPG1 adjuvant groups induce high IFN-γ levels, which are consistent with previous reports [47]. IL-4 is a cytokine with pleiotropic functions and plays an important role in Th2 responses. High levels of IL-4 can help clear the influenza virus, combat excessive inflammation, and promote recovery [48]. In this study, we detected high levels of IL-4, which may be consistent with higher levels of humoral immunity to some extent. We also compared the effect of doses on the immunity of the nanoparticle vaccine. There was no significant difference in antibody levels between the 10 µg group (data not shown) and the 5 µg group. Therefore, 5 µg was chosen in this study.

After the challenge, the weight changes of the mice were recorded for 14 days. According to previous studies, mice showing >25% of their initial body weight loss were defined as an end-point and were euthanized [49,50]. In this study, two mice died in the PBS-group challenge with the H1N1 virus, and three mice lost more than 25% of their initial body weight during the observation period. Although the weight of the mice recovered from the ninth day gradually, the body weight did not return to the pre-challenge level (approximately 83.39% for H1N1 and 97.13% for H3N2) until the 14th day. The results further demonstrated the serious hazard of SIV; although the fatality rate is not high, SIV infection causes progressive wasting, poor growth performance, and may cause death from co-infection with other respiratory pathogens or bacterial diseases. Therefore, the clinical infection and morbidity of swine flu should arouse attention.

So far, at least seven different serum subtypes of SIV have been found, and classical swine H1N1, avian-like H1N1, and humanoid H3N2 strains are widely prevalent in pigs. Serological epidemiological investigation of SI showed that the distribution and evolution of SIV in China is becoming increasingly complex [51]. The application of BNV is of great significance to block the transmission of swine influenza virus in China.

Although mice are often used as animal models for influenza vaccine evaluation [52], there are species differences between different animals. The limitation of this study is that the bivalent nanoparticle vaccine (BNV) has been verified only in mice, and the immune protective effect in pigs needs to be further verified.

## 5. Conclusions

Our data provides insights into a bivalent nanoparticle swine influenza vaccine generated by the baculovirus-insect cell system. The BNV contains modified HA proteins and can induce robust immunity in mice. MF59 and CPG1 adjuvants can remarkably enhance the HAI antibody titers of BNV. The MF59 adjuvant showed a balanced Th1/Th2 immune response, and the CPG1 adjuvant tended to show a Th1-favored response. The BALB/c challenge test showed that BNV could provide adequate protection. This study provides a candidate vaccine for the prevention and control of swine influenza. However, immune efficacy in pigs needs to be further evaluated.

## Figures and Tables

**Figure 1 viruses-14-02443-f001:**
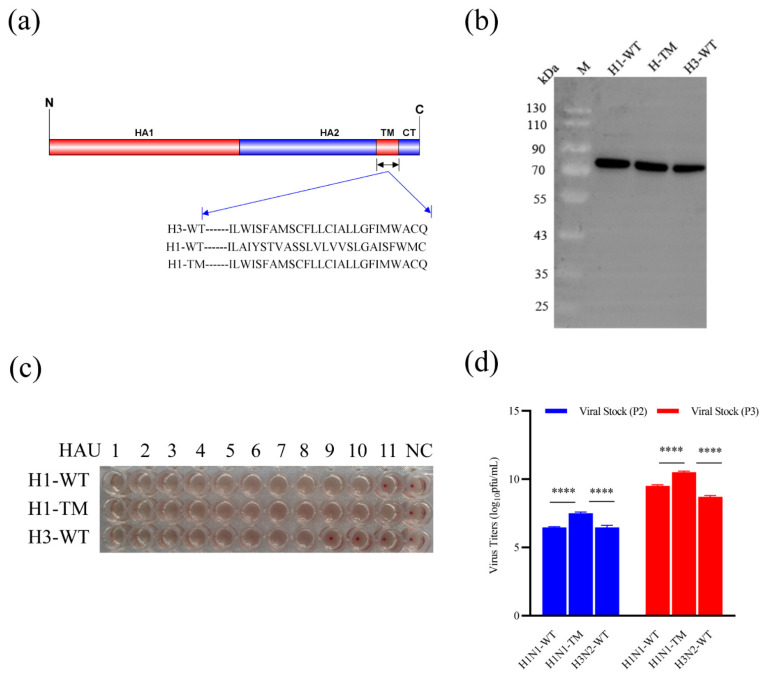
Recombinant baculovirus production and characteristic analysis. (**a**) Structural schematics of SIV HA proteins, where the amino acid sequences of the TM domain are shown. H3-WT denotes H3 wild-type (WT) HA protein; H1-WT denotes H1 wild-type HA protein; and H1-TM denotes H1 wild-type HA protein with the replaced H3 TM. Amino acid sequences of H3-WT, H1-WT, and H1-TM were aligned and designated. (**b**) Western blots of H1-WT, H1-TM, and H3-WT protein expression. The molecular weight of HA was approximately 70 kD. (**c**) Hemagglutination assays of H1-WT, H1-TM, and H3-WT protein. PBS with 0.5% chicken erythrocytes was included as a negative control (NC). All experiments were performed 3 times. (**d**) Viral titers of H1-WT, H1-TM, and H3-WT recombinant viruses of different generations of viral stocks (**** *p* < 0.0001).

**Figure 2 viruses-14-02443-f002:**
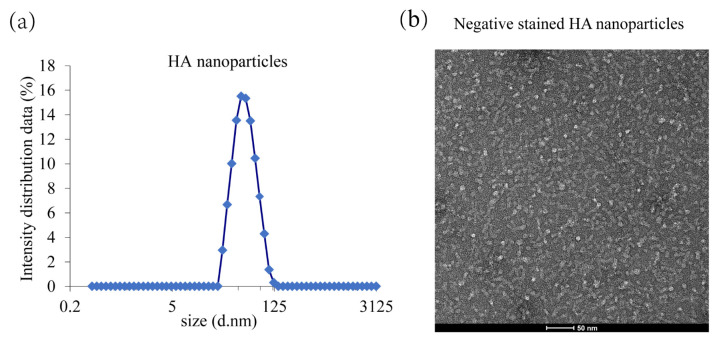
Dynamic light scattering (DLS) and transmission electron microscopy (TEM) analysis of HA nanoparticle. (**a**) DLS analysis of HA nanoparticle. (**b**) TEM analysis of HA nanoparticle.

**Figure 3 viruses-14-02443-f003:**
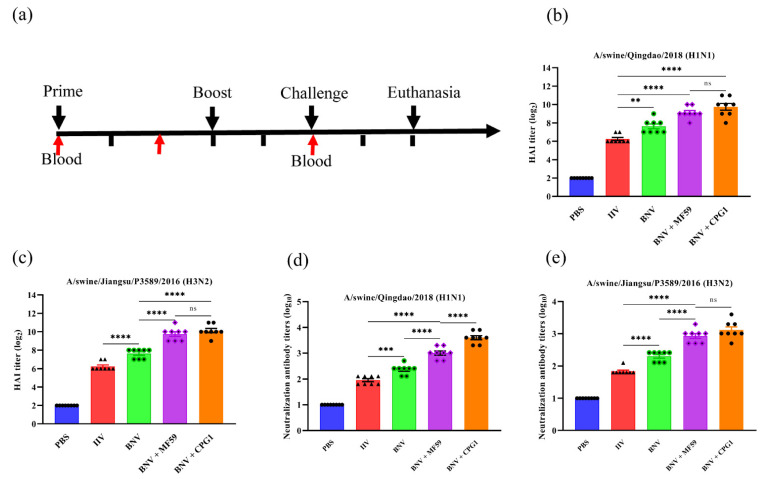
HAI and VN antibody responses in BNV immunized mice. (**a**) Schematic flow diagram of the animal immunization and challenge. (**b**,**c**) HAI antibody titers in mice after immunization with swine influenza bivalent nanoparticle vaccines (*n* = 8). The results were calculated by the value of log2. Each symbol represents an individual mouse, and the horizontal bar indicates the geometric mean of the group. (**d**,**e**) VN antibody titers in mice after immunization with swine influenza bivalent nanoparticle vaccines (*n* = 8). The results were calculated by the value of log10. Each symbol represents an individual mouse, and the horizontal bar indicates the geometric mean of the group. Values are presented as mean ± standard deviation (S.D). Statistical significance analysis was performed using one-way ANOVA and then multiple comparison tests (** *p* < 0.01; *** *p* < 0.001; **** *p* < 0.0001; ns, *p* > 0.05), and n.s. indicates no significance between the two compared groups (*n* = 8). Three parallel experiments in each group were conducted.

**Figure 4 viruses-14-02443-f004:**
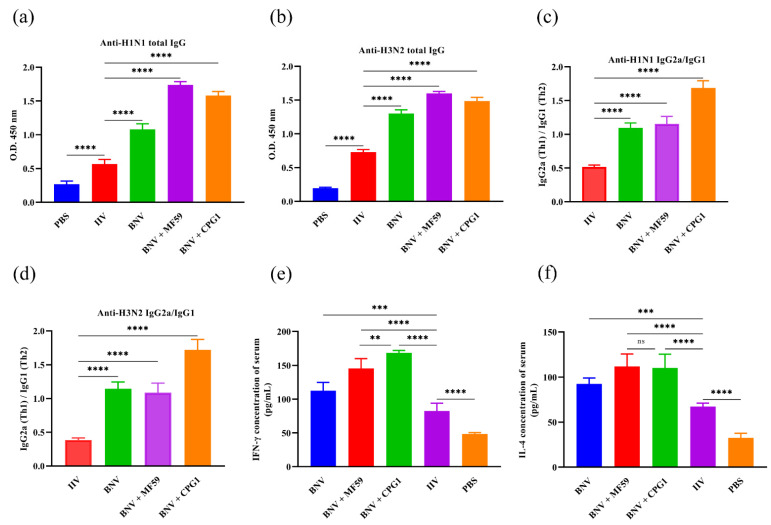
Virus-specific IgG ELISA and cytokine analysis in BNV immunized mice. (**a**,**b**) Swine influenza H1N1 and H3N2-specific IgG titers in serum two weeks after the final immunization. The minimum detectable dose of mouse IgG is less than 1.0 µg/mL. (**c**,**d**) Levels of IgG1 and IgG2a in serum of immunized mice were determined using ELISA two weeks after boost immunization. The minimum detectable dose of mouse IgG1 and IgG2a is less than 1.0 µg/mL and 0.1 µg/mL, respectively. (**e**,**f**) *IFN-γ* and *IL-4* levels in serum from immunized mice were analyzed by mouse *IFN-γ* and *IL-4* enzyme immunoassay kit. The minimum detectable dose of mouse *IFN-γ* and *IL-4* is typically less than 1.0 pg/mL. Values are presented as mean ± standard deviation (S.D). Statistical significance analysis was performed using one-way ANOVA and then multiple comparison tests (** *p* < 0.01; *** *p* < 0.001; **** *p* < 0.0001; ns, *p* > 0.05), and *n.s*. indicates no significance between the two compared groups (*n* = 8). Three parallel experiments in each group were conducted.

**Figure 5 viruses-14-02443-f005:**
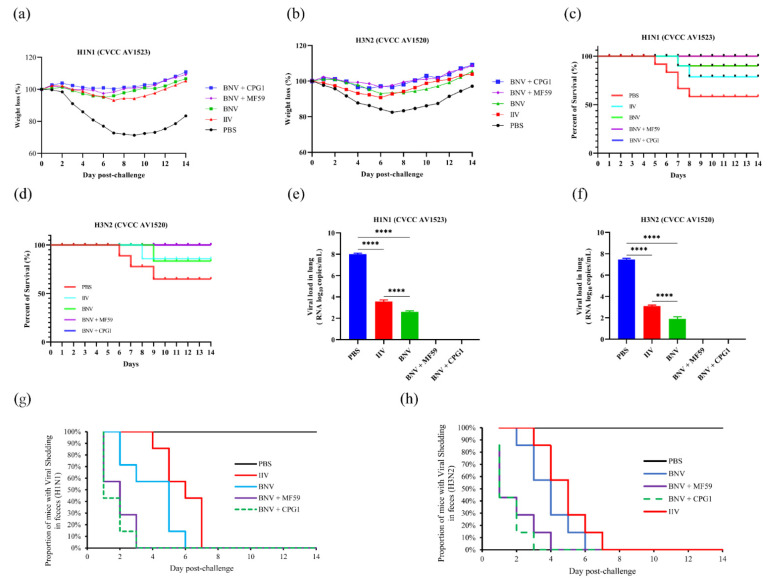
Mice were immunized with BNV-nanoparticle, MF59-adjuvanted, or CPG1-adjuvanted BNV and challenged with swine influenza H1N1 (CVCC AV1523) and H3N2 (CVCC AV1520) viruses. (**a**,**b**) Weight changes of mice challenged with swine influenza H1N1 and H3N2 in each group are shown. Values indicate the mean weight changes of all mice in each group after the virus challenge. (**c**,**d**) The survival rates were analyzed by Kaplan–Meier methods (*n* = 8/group). (**e**,**f**) Viral loads in lung tissues of mice six days post-challenge with H1N1 and H3N2 viruses are shown (**** *p* < 0.0001). (**g**,**h**) Duration of viral shedding in feces samples post-challenge with H1N1 and H3N2 viruses.

**Figure 6 viruses-14-02443-f006:**
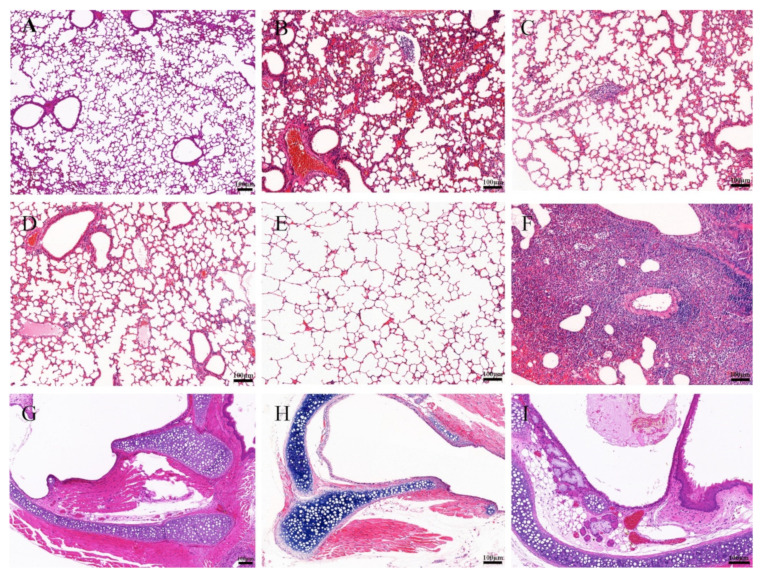
Histopathologic examination of lung and trachea lesions in mice at sixth day post-challenge of H1N1 (CVCC AV1523). (**A**–**F**): Lung changes of different groups. (**A**): NC (lung); (**B**): inactivated vaccine (IIV) group; (**C**): BNV group; (**D**): BNV + MF59 group; (**E**): BNV + CPG group; (**F**): PBS control group; (**G**–**I**): trachea changes of different groups; (**G**): NC (trachea); (**H**): inactivated vaccine (IIV) group; (**I**): BNV group (Scale bar = 100 µm).

**Figure 7 viruses-14-02443-f007:**
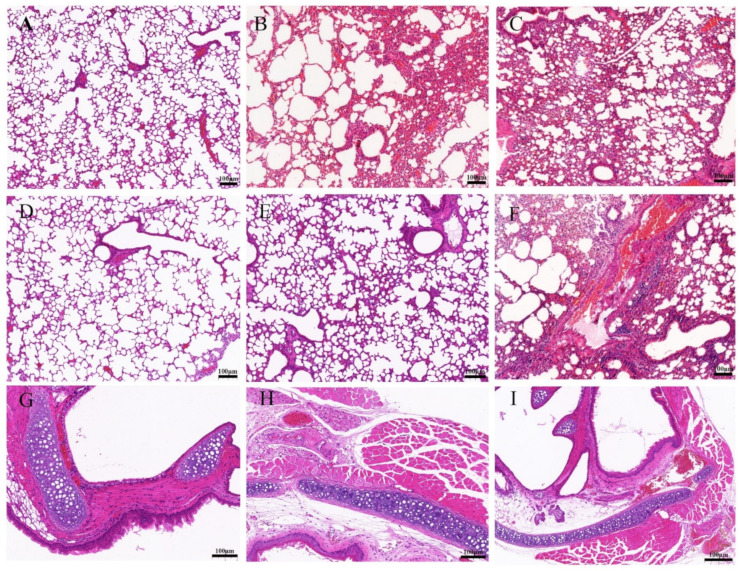
Histopathologic examination of lung and trachea lesions in mice at sixth day post-challenge of H3N2 (CVCC AV1520). (**A**–**F**): Lung changes of different groups. (**A**): NC (lung); (**B**): inactivated vaccine (IIV) group; (**C**): BNV group; (**D**): BNV + MF59 group; (**E**): BNV + CPG group; (**F**): PBS control group; (**G**–**I**): trachea changes of different groups; (**G**): NC (trachea); (**H**): inactivated vaccine (IIV) group; (**I**): BNV group (Scale bar = 100 µm). All specimens were processed according to the SOP procedure for pathological examination. Dehydration, pruning, embedding, sectioning, staining, sealing, and microscopic examination.

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
