# Peer review of "Nanoparticle-Based Bivalent Swine Influenza Virus Vaccine Induces Enhanced Immunity and Effective Protection against Drifted H1N1 and H3N2 Viruses in Mice"

_viruses, 2022, doi:10.3390/v14112443_

Round 1

Reviewer 1 Report

MATERIAL AND METHODS

It is missing a chapter describing the cell lines that were used in the study.

HANP should be written in full when firstly cited. It would be interesting to include a schematic figure showing the HA nanoparticles.

Line 96: Centrifugation speed should be written in g instead rpm.

The synthesis of HANP should be better described.

Lines 116-117: BNV was co-administered “ with 5 μg MF59 or 5 μg CPG1” instead “5 μg MF59 and 5 μg CPG1”. This comment is because it seems that animals had been vaccinated with both adjuvants instead one or another.

What are the strains contained in influenza commercial vaccine?

Lines 125- Why the viruses used for the HAI were propagated in insect cells instead more canonic cell cultures such as MDCK cells ?

IVC should be written in full when firstly cited.

Line 134: “with or without MF59 or CPG1 adjuvant”  instead “with or without MF59 and CPG1 adjuvant”.

Lines 135-136: What was the immunization protocol used to vaccinate the mice with the commercial inactivated vaccine. How much protein the animals received?

Line 139: “of swine influenza H1N1 or H3N2” instead “of swine influenza H1N1 and H3N2”.

Influenza is mainly a respiratory virus, why to assess the viral RNA in feces?

Regarding the HAI assay, it seems that one step in the experimental protocol is missing. For instance, the removing of nonspecific agglutinins from the serum samples by adding red blood cells to the RDE-treated was not described in the text.

Line 164: ...neutralization titers are expressed as the highest dilution of sera. This sentence is incomplete. What was the endpoint? 100% of inhibition?

Line 176: “euthanized” instead “sacrificed”

RESULTS

Figures  3 and 4: The quality of the images is very poor. Regarding the figure 3,it is strongly suggested split it in two. Perhaps, the panels 3f-k could be depicted in another figure.

Figures 3B and 3C. Because the serum dilution was performed in log2, it is suggested to change the Y axis of the graphic, to better represent 2 fold dilutions.

Line 233 “...with BNV with or without MF59 or CPG1 adjuvant” instead “with BNV with and without MF59 and CPG1 adjuvant”

The authors used rBV supernatants as antigen to perform the hemagglutinin inhibition assay. This choice do not seems to be the better one. Why not use de corresponding influenza virus as antigen?

Figure 3d, the authors state that there are no significant differences in neutralization antibody titers in mice immunized with MF59 or CPGI adjuvants. However, the data displayed in the graphic showed that using CPGI as adjuvant elicited higher neutralization antibody titers.

Figure 3j and 3k the color of the bars of each experimental group as well as their order in the graphic should be the same in all graphics.

Lines 285-286. The sentence: “We next determined the protection level of BNV against homologous but drifted virus attacks” was poorly written.

Figures 4C and 4D there are no comments about the results depicted in that figures.

Figure 4g and 4h, the authors argue that there was virus shedding in feces. However, they only assess this shedding by real time PCR. In order to prove that there as truly a viral shedding, it should be necessary to perform another kind of experiment, such as viral titration.

According to the authors, ‘No obvious pathological changes were observed in mice vaccinated with BNV and BNV plus MF59 or CPG1 adjuvant groups”. However the histopathologic images depicted in figures 5A and B suggested that there was an inflammatory process in mice vaccinated according to all the protocols. It is strongly suggested to show the results also in graphics depicting the inflammation score, such as neutrophils infiltration, epithelial desquamation, for example and overall histopathological score.

Line 355 IBEVS should be written in full when firstly cited.

Discussion

The authors detected high levels of IL-4. This cytokine seems to play a very controversial role during influenza infection. The authors should discuss their findings.

Line 356: The authors state that nanoparticle vaccine expressed by IBEVS and fully protected against drifted SIVes. In spite the promising results, some animals that were immunized with BNV died.

Lines 389-390: The authors state that other nanoparticles described above elicited mainly cellular immune responses, whereas the nanoparticle studied in this manuscript elicit balanced humoral and cell mediated immune responses. Therefore, the authors should include references to corroborate this assumption.

Conclusions: The conclusions are completely out of the range of their results and the scopus of their study They should be carefully rewritten.

Reviewer 2 Report

Initially, I would like to congratulate the authors for the excellent manuscript and the importance of the results for the production of a vaccine that shows promise in the fight against swine flu.

In my opinion, the authors should add:

a) the limitations of the study in the discussion

b) what would be the economic impact of this vaccine when compared to vaccines that use embryonated chicken eggs?

c) What are the most important points that would justify the choice of the bivalent nanoparticle vaccine (BNV)?

Reviewer 3 Report

This manuscript studied bivalent nanoparticle SIV vaccine containing modified HA proteins. The authors evaluated the protective efficacy of a recombinant, baculovirus-insect cell system expressed bivalent nanoparticle in mice challenged with drifted swine influenza H1N1 and H3N2 viruses. Overall, the data/analysis presented high levels of HAI antibodies, neutralization antibodies, and antigen-specific IgG antibodies were induced. However, the study design, method, and analysis have not been carefully presented and reasonably interpreted. In particular, the evaluation of T cell immune response was not comprehensive.  There are a few concerns listed below that would be important to address.

1. The quality of the Figures in this manuscript is poor. Please provide better resolution figures.

2. Please provide the information on hemagglutination assay in Materials and Methods

3. The color of the plots is not consistent across the manuscript. Please revise all the bar plots in this manuscript with the same graph type and color, as well as provide the limit of detection in all the bar plots.

4. There are two Figure 5 in the manuscript. Please arrange the plots and move the unnecessary plots to supplemental.

5. How many para-level experiments were performed in this study? I did not find the information in the figure legend or method.
